# Circulating Tumor Cells Develop Resistance to TRAIL-Induced Apoptosis Through Autophagic Removal of Death Receptor 5: Evidence from an In Vitro Model

**DOI:** 10.3390/cancers11010094

**Published:** 2019-01-15

**Authors:** Julianne D. Twomey, Baolin Zhang

**Affiliations:** Office of Biotechnology Products, Center for Drug Evaluation and Research, Food and Drug Administration, Silver Spring, MD 20993, USA; Julianne.twomey@fda.hhs.gov

**Keywords:** circulating tumor cells, CTCs, breast cancer, metastasis, death receptor, TRAIL, apoptosis, in vitro model

## Abstract

Circulating tumor cells (CTCs) in the peripheral blood are the precursors to distant metastasis but the underlying mechanisms are poorly understood. This study aims at understanding the molecular features within CTCs, in relation to their metastatic potential. Using in vitro CTC models, in which breast cancer cell lines were cultured in non-adherent conditions simulating the microenvironment in the blood stream, we found that the suspension culture resulted in resistance to TNF-related apoptosis inducing ligand (TRAIL)-mediated cell death. Such a resistance was directly correlated with a reduction in surface and total levels of DR5 protein. In the non-adherent state, the cells underwent a rapid autophagic flux, characterized by an accumulation of autophagosome organelles. Notably, DR5 was translocated to the autophagosomes and underwent a lysosomal degradation. Our data suggest that CTCs may evade the TNF cytokine-mediated immune surveillance through a downregulation of the death receptor (DR) expression. The data warrants further studies in cancer patients to find the status of DRs and other molecular features within primary CTCs, in relation to disease progression or chemoresistance.

## 1. Introduction

Circulating tumor cells (CTCs) are shed into the vasculature from the primary tumor and circulate within the bloodstream. CTCs have the potential to seed in distant organs in the body (e.g., bone, liver, lymph nodes, and lungs), where they can grow into new tumors and further disseminate CTCs. Several clinical trials have shown that CTC presence in the blood is a surrogate of the risk of progression or death in patients with metastatic solid tumors, including breast cancer [1,2,3]. CTC status is also being explored as a “liquid biopsy”, for real-time monitoring of patients’ response to therapy [4,5]. Reports from different groups, however, have shown considerable variation in CTC detection rates and relationship with prognosis or therapy response [6,7,8,9], thus limiting their clinical utility as a routine diagnostic tool. At present, a major challenge in the field is to find the molecular signatures within CTC subsets that drive tumor metastasis or resistance to therapy. 

CTCs in the peripheral blood are constantly exposed to a perilous environment—a lack of adherent matrix, high fluid shear stress, and cancer targeting immune cells and cytokines [10,11,12]. The metastatic process is highly vulnerable to these stress signals, with the destruction of most CTCs within minutes by anoikis—a form of apoptosis induced by loss of cell–cell or cell–matrix adhesion [13,14]. In a rat model of CTCs, only a small proportion of the anoikis resistant population (~0.1%) progressed to form micrometastases [15]. Activation of autophagy signaling within CTCs has been proposed as a mechanism through which these cells survive the cumulative stress signals (e.g., detachment, hypoxia) in the circulation, thereby contributing to their metastatic potential [13,16,17]. Autophagy is characterized by the formation of an autophagosome that can engulf the cytoplasmic components, as well as selective targets that are tagged with ubiquitin-binding proteins, such as p62 [18], which are degraded following lysosome fusion [19]. Cancer cells can utilize pro-survival autophagy to resist chemotherapies [20], radiation [21], and targeted therapy [22]. This process is tightly regulated by a complex signaling network that involves beclin-1, microtubule-associated protein 1a/1b-light chain 3 (LC3), ATG7, and RAB7/9, as well as other autophagy related proteins.

CTCs’ metastatic potential also depends on their susceptibility to immune cells (e.g., macrophage, natural killer cells) and circulating cytokines; including Fas ligand (FasL), tumor necrosis factor α (TNF-α), and TNF-related apoptosis inducing ligand (TRAIL). These cytokines, named death ligands, can induce apoptosis or necroptosis through binding and activation of their cognate death receptors (DRs) Fas, TNF receptor 1 (TNFR1), or DR4/DR5 in target cells, thereby serving a critical mechanism in immunosurveillance and cancer immunity. In many cancer cells, however, activation of this process is hindered due to molecular alterations at several signaling checkpoints, including a deficiency in the expression of DRs on the cell surface. Our lab has previously shown that DR4/DR5 undergo rapid endocytosis and, in some cases, are sequestered into nuclei and autophagosome structures in breast cancer cells [23,24]. Other factors include overexpression of anti-apoptotic proteins (e.g., IAPs, FLIP, Bcl-2) [25] or enhanced cell survival signaling, via the PI3K/Akt pathway [26]. The observations were made exclusively in primary tumor cells or tumor cell lines, but little is known about the status of the DR apoptosis signaling components in CTCs.

CTCs are extremely rare events with a typical frequency of one CTC per billion leukocytes [27], a confounding factor in CTC isolation and functional characterization. To date, the CellSearch CTC detection system is the only FDA-approved test for clinical use [28]. The method is designed to capture the circulating cells of epithelial origin (CD45-, EpCAM+, and cytokeratins 8, 18+, and/or 19+), in the whole blood [2,3,8]. However, CTCs are highly heterogenous and the surface markers used in the CellSearch assay are not always present in all CTC populations. Attempts are being made to develop new CTC isolation platforms [29,30,31] or in vitro CTC models [32,33,34,35,36]. In this study, we developed a long-term suspension cell culture model using human breast cancer cell lines, simulating the peripheral blood environment. This model allows a direct comparison between CTCs and their counterpart primary tumor cells being cultured under adherent conditions. Similar approaches have previously been used to examine subpopulations in cancer cell lines [32], death receptor surface localization [33,34], and chemotherapy resistance [35]. Using in vitro CTC models, we found that breast cancer cells in suspension rapidly undergo autophagic flux, resulting in a degradation of DR5 and resistance to TRAIL-mediated apoptosis. These data suggest that CTCs may survive the TNF cytokine-mediated killing through a selective downregulation of death receptors in the circulation environment.

## 2. Results

### 2.1. Breast Cancer Cells Develop Resistance to Recombinant Human (rh)TRAIL Induced Apoptosis under Non-Adherent Culture Conditions

We aimed to unravel the molecular mechanisms through which CTCs adapt to evade the TNF cytokines in the blood stream, including TNFα, FasL, and TRAIL. To this end, we developed an in vitro CTC model, wherein human cancer cell lines were cultured in non-adherent conditions, simulating the circulating microenvironment. As controls, the same cell lines were also grown under monolayer adherent conditions. Briefly, a panel of three human breast cancer cell lines (MDA-MB-231, MCF7, and ZR75-1) were grown in suspension or monolayer for seven days. Suspension cultured cells aggregate in either a stellate formation, such as the MDA-MB-231 and ZR75-1 cell lines, or in spheroids, such as the MCF7 cell line (Appendix A). Following the seven days of suspension culture, cells were non-enzymatically dissociated and re-seeded into suspension or monolayer conditions within 96 well plates. Cells were treated with recombinant human (rh)TRAIL at two dosage levels over 24 h, and measured for viability, using an MTT assay (Figure 1a). rhTRAIL concentrations reflected the known IC50 concentrations, for each breast cancer cell (BCC) line from previous studies [37]. Significant differences in viability between the suspension and the monolayer cultures were seen following six hours of treatment in the MDA-MB-231 cells under high dosage (50 ng/mL; OD 0.41 ± 0.02 monolayer versus OD 0.84 ± 0.01 suspension condition, *p* < 0.001), and by twelve hours of treatment under low dosage (5 ng/mL; OD 0.60 ± 0.02 monolayer versus 0.76 ± 0.02 suspension condition, *p* = 0.007). rhTRAIL induced cytotoxicity in the monolayer-cultured MDA-MB-231 cells in a time-dependent manner, resulting in a 24% (OD 0.24 ± 0.02) relative viability at 24 h of incubation at the concentration of 50 ng/mL. In contrast, the MDA-MB-231 cells cultured in suspension conditions underwent an initial reduction in viability, which was then maintained around 60%, following at 24 h of incubation (OD 0.62 ± 0.01, *p* = 0.007) (Figure 1A). Similar results were seen by 9 h of rhTRAIL incubation in the ZR75-1 cells (OD 0.71 ± 0.02 monolayer versus OD 0.89 ± 0.06 suspension condition, *p* = 0.05) at 50 ng/mL and MCF7 cells (OD 0.78 ± 0.02 monolayer versus OD 0.91 ± 0.02 suspension condition, *p* = 0.011) at 1000 ng/mL. Suspension cultured cells maintained a higher cell viability, compared to monolayer cultures, at 24 h of treatment, for the ZR75-1 cells (OD 0.37 ± 0.5 monolayer versus 0.70 ± 0.01 suspension condition, *p* = 0.003) and the MCF7 cells (OD 0.65 ± 0.2 monolayer versus OD 0.89 ± 0.01 suspension, *p* = 0.001). The delayed apoptosis execution was also shown in the western blot analysis (Figure 1b). rhTRAIL treatment induced poly (ADP-ribose) polymerase (PARP) and caspase 3 and 8 cleavage after one hour, in monolayer-cultured cells, compared to three hours in the suspension-cultured MDA-MB-231 cells, four hours in ZR75-1 cells, and nine hours in the MCF7 cells. As the MCF7 cells are deficient in caspase 3 [38], the activation of the extrinsic apoptotic signaling pathway might include a compensatory activation of the effector caspases-6 or -7, resulting in a cleavage of PARP.

### 2.2. Non-Adherent Culture Decreases the DR5 Surface and Total Protein Expression

We have previously shown that breast cancer cellular sensitivity to TNF death ligands is correlated with the corresponding death receptor (DR) expression on the plasma membrane [23,37]. To test this possibility in the non-adherent cultured cells, we performed flow cytometry analysis using antibodies specific to DR4, DR5, Fas, and TNFR1, respectively (Figure 2a). Surface expression of DR5, Fas, and TNFR1 was detected in all monolayer-cultured cells for the MDA-MB-231, ZR75-1, and MCF7 cell lines. Following the suspension culture, DR5 surface expression was significantly reduced. By contrast, DR4, TNFR1, and Fas did not show significant changes following suspension culture, except for Fas in the ZR75-1 cells (Appendix A). Though changes of the DR4 surface expression were below the level of detection within our experiments, even low-level changes of DR4 might contribute to TRAIL-resistance due to apoptotic signaling capability upon TRAIL-binding [39]. We also assessed the expression of other surface receptors, including an HLA-Class I Major Histocompatability Complex (MHC), decoy receptors 1 (DcR1) and 2 (DcR2), integrin β1 (ITGβ1), and EGFR (Appendix A). Overall, these receptors were not consistently affected in the suspension culture across the cell lines.

Next, we measured the total protein levels of individual DRs using immunoblot analysis. DR5 and TNFR1 were both significantly reduced over the seven days of suspension culture, with a gradual decrease in the total DR5 expression seen in the MDA-MB-231 and the ZR75-1 cells and a more rapid decrease in the MCF7 cells (Figure 2b, Appendix A). DR4 and Fas expression did not show a consistent expression profile over the time or across the samples (Appendix A). The expression of DR4 did demonstrate a minor reduction, while not significant, which could contribute to the acquired resistance to TRAIL-induced apoptosis. At the transcriptional level, mRNA expression of all four DRs did not show significant changes in the two cell lines cultured in suspension, except for the MDA-MB-231 cells, which showed a slight decrease in the DR4 expression (Table 1). These data showed that the reduction in DR protein expression likely involves post-translational mechanisms.

### 2.3. Autophagic Flux is Upregulated in Cells Cultured in Non-Adherent Conditions

Previous studies in our lab have shown that autophagic induction causes breast cancer cells to internalize death receptors leading to TRAIL resistance [24]. To examine the status of autophagy, cell samples from suspension culture were analyzed for the autophagy markers of LC3 and the ubiquitin-binding cargo protein p62 (Figure 3a). In suspension culture, all cell lines displayed a significant increase in LC3-II/LC3-I ratio, and a continual decrease of p62 expression. The formation of autophagosomes was confirmed using multispectral imaging flow cytometry analysis (see details in Methods). 

The levels of autophagosomes were found to be considerably higher in the cells cultured under suspension, compared to the respective monolayer cultures, in all three cell lines (Figure 3b). Monolayer cultures displayed around 2–6% of the cell population having more than seven LC3 puncta per cell, which was determined to be the basal level of autophagy in the monolayer culture (Appendix A). There was an increase in this population to around 19–25%, following 7 days of suspension culture (Figure 3b: MDA-MB-231: 5.7 ± 0.4% to 24.5 ± 1.8%, *p* < 0.01; ZR75-1:4.0 ± 0.1% to 18.6 ± 4.1%, *p* < 0.01; MCF7: 1.9 ± 1.1% to 20.9 ± 2.6%, *p* < 0.01).

Further, we analyzed the co-localization of the LC3-positive puncta with lysosomal-associated membrane protein 1 (LAMP-1) puncta, using bright detail intensity similarity analysis (BDS) for the LC3-AF488 and the LAMP1-PE. LAMP-1 is a transmembrane glycoprotein that exists in lysosomal membranes. The population of cells that were undergoing autophagic flux (BDS > 1.8) increased from the monolayer (MDA-MB-231: 1.2 ± 0.6%; ZR75-1: 2.0 ± 0.2%; MCF7: 1.6 ± 0.8%) to the suspension culture (MDA-MB-231: 27.3 ± 4.8%; ZR75-1: 13.1 ± 3.7%, *p* < 0.05; MCF7: 32.1 ± 6.4%, *p* < 0.01). These data show that autophagic flux is activated in the non-adherent-cultured cells, displaying sustained autophagosome and autolysosome vesicles.

### 2.4. DR5 is Localized to Autophagosomes for Degradation

We have previously shown DR5 translocation to the autophagosomes in cancer cells under normal growth conditions [24]. To examine this possibility in the non-adherent cultured cells, a co-localization study was performed between DR5-PE and LC3-AF488. Consistent with the immunoblot data, surface levels of DR5 were decreased and the double-stained cell population was reduced (Figure 4a). The co-localization quantification was performed on the dual-positive population. Following the suspension culture, the bright detail similarity significantly increased in all three cell lines, with the DR5 staining transitioning from diffuse to more punctate over the course of seven days. This was further confirmed with confocal microscopy on the ZR75-1 cells, grown in monolayer or suspension conditions (Figure 4b). Collectively, the data suggest that the reduction in the DR5 protein level is likely through a localization to the autophagosome, and a subsequent degradation.

## 3. Discussion

As an essential step in the metastatic cascade CTCs must survive the immune defense system involving circulating immune cells and cytokines, such as TNF-α, FasL, and TRAIL [40]. Here, we present a novel mechanism through which the CTC subsets may evade the TNF cytokine-mediated toxicity effects. Using in vitro CTC models, we found that the CTC-like cells undergo a rapid autophagy flux, which targets death receptors (DRs), especially DR5, for autolysosome degradation. The resultant cells become resistant to TRAIL-induced apoptosis.

Previous studies have utilized the suspension culture of cancer cell lines to analyze how CTCs undergo changes in cell proliferation and metabolic pathways [41], as well as in drug sensitivities [42,43]. In this study, we tested three breast cancer cell lines (MDA-MB-231, ZR75-1, and MCF7) with a known difference in their sensitivity to TNF death ligands under monolayer culture [44]. When cultured in a non-adherent condition, all three cell lines showed a delayed apoptotic response to the rhTRAIL, with a significant proportion of cells surviving the cytokine-mediated killing. This was correlated with a decrease in the DR5 protein expression (Figure 4a,b). CTCs have shown reduced adhesion receptors, such as E-cadherin, as these are not needed within the circulation [32,36]. It is well documented that matrix detachment increases cellular stress to trigger autophagy in cancer cells [16,17]. As expected, the non-adherent cells displayed rapid autophagic flux, as shown by the distinct autophagy markers (Figure 3a). Notably, DR5 was found to be co-localized in the autophagosome organelles. DR5 mRNA level was not affected; thus, the loss of DR5 protein expression could be attributed to an autolysosomal degradation. As shown in the monolayer cancer cells [23,37], a deficiency in the cell surface DR5 is correlated with a resistance to TRAIL-induced apoptosis.

Our findings also have a potential implication in understanding tumor resistance to DR-targeted therapies. Due to its proapoptotic activity, DR5 is an attractive target for cancer treatment. There have been several clinical trials testing recombinant human TRAIL and agonistic anti-DR5 monoclonal antibodies. However, these agents have only shown limited clinical efficacy in the completed trials [45,46,47,48]. Our data suggest that CTCs may be targeted within the blood stream by endogenous or introduced TRAIL or anti-DR5 molecules, resulting in a DR5 deficiency on the cell surface. In support of this notion, we have previously shown that rhTRAIL or anti-DR5 antibody induces an internalization of the DR5 from the plasma membrane of the monolayer cancer cells [23,37]. The internalized DR5 can be processed through an autophagic degradation in the circulating cells. The resulting CTCs will lose the susceptibility to those agents, thus resulting in a treatment failure. Such a resistance mechanism may be challenged using combination therapy with autophagy inhibitors [24,49,50], in light of the evidence that autophagy blockade restores surface DR5 expression, thereby, augmenting the TRAIL-induced apoptotic signaling [24]. It should be mentioned that our results did not discount the possibility of a contribution of the DR4 to the acquired resistance to TRAIL-induced apoptosis, which we have seen here. TRAIL-induced apoptosis was found to occur preferentially through DR4, in a number of cell lines [39,51], which might be impacted by a non-adherent environment.

In conclusion, we have shown that autophagic flux may be a mechanism through which tumor cells internalize and degrade death receptors when they are shed from the primary tumor and circulate in the blood stream. The data warrants further studies in cancer patients to find and confirm the status of DRs and other molecular features within the primary CTCs, in relation to disease progression or chemoresistance. The acquired information can enhance the clinical utility of CTC detection to promote personalized cancer medicine.

## 4. Materials and Methods

### 4.1. Cell Lines and Culture Conditions

Human breast cancer cell lines of MCF7 (ATCC, HTB-22), ZR75-1 (ATCC, CRL-1500), and MDA-MB-231 (ATCC, HTB-26) were obtained from the American Type Culture Collection (Manassas, VA, USA) and sub-cultured, as per supplier’s instructions. MCF7 cells were cultured in Eagle’s Minimum Essential Medium (EMEM) (ATCC) supplemented with 10% fetal bovine serum (FBS) (Corning, Corning, NY, USA), and Human Insulin (Invitrogen, Carlsbad, CA, USA). ZR75-1 cells were cultured in the RPMI-1640 Medium (ATCC, 30-2001), with 10% FBS, and the MDA-MB-231 cells were cultured in the DMEM/F-12 (1:1) medium (Mediatech, Herndon, VA, USA), containing 5% FBS, 4 mM glutamine, 50 µM β-Mercaptoethanol, and 1 mM sodium pyruvate. Cell lines were maintained at 37 °C with 5% CO_2_. For the suspension culture, cells were seeded into Corning Ultra-Low Attachment plates, which inhibits immobilization to the surface. Cells were maintained in suspension with media changes every other day, for up to seven days.

### 4.2. TRAIL Cytotoxicity Assay

Cells were seeded into monolayer or suspension for 7 days, at which point the cells were isolated, using a non-enzymatic dissociation buffer (CellStripper, Invitrogen, Carlsbad, CA, USA), and re-seeded into either the tissue culture polystyrene or the Ultra-Low Attachment 96 well plates, at a concentration of 10,000 cells/well. Cells were maintained in a complete medium, overnight, prior to the TRAIL treatment. The cells were treated with the recombinant human TRAIL (TRAIL) (R&D Systems, 375-TEC), at two different doses (10 ng/mL and 100 ng/mL for MDA-MB-231 and ZR75-1 cell lines; 50 ng/mL and 500 ng/mL for MCF7 cell lines). The concentrations were based on our previous research, in which the IC50 for TRAIL treatment over 24 h was found to be 1–5 ng/mL for MDA-MB-231, 7–8 ng/mL for ZR75-1, and >500 ng/mL for MCF7 cells [37]. Plates were incubated with TRAIL over 24 h and analyzed for either viability or protein expression, at indicated timepoints.

### 4.3. Cell Viability Assays

Cell viability assays were conducted to determine the effect of TRAIL on the cells cultured in the monolayer and the suspension conditions. A standard colorimetric MTT assay was performed to determine the percentage of the viable cells. 100 μL of 3-(4,5-dimethylthiazol-2-yl)-2,5-diphenyltetrazoilum bromide (Thiazolul Blue Tetrazolium Bromide; 2 mg/mL in PBS) was added to the culture and incubated with the cells, for 45 min. Cells were then spun down for 10 min, at 1500 rpm and the MTT solution was removed. The resulting crystals were solubilized with DMSO and the absorbance was measured at 562 nm, using a microplate reader. Absorbance was normalized with respect to the non-treated control, at each time-point, and the viability was reported as percentages.

### 4.4. Flow Cytometry Detection of Surface Death Receptors

The expression levels of death receptors DR4, DR5, Fas, and TNFR1 were measured on cells cultured in either the monolayer or the suspension culture, for seven days. Cells were dissociated into single cell suspensions, using a non-enzymatic dissociation buffer (Cell Stripper). Cells were spun down for 5 min at 800× g and re-suspended at 5.0 × 10^6^ cells/mL, in a blocking solution (1.0% bovine serum albumin, 5.0% normal goat serum (Invitrogen), in PBS). Cells were blocked against the non-specific binding for 20 min on ice. Cells were then labeled with anti-DR4-PE (IgG_1_, R&D Systems, FAB347P), anti-DR5-PE (IgG_2B_, R&D Systems, FAB6311P), anti-TNFR1-PE (IgG_1_, R&D Systems, FAB225P), or anti-Fas-PE (IgG_1κ_, BD Pharmigen, 555674), for 45 min, in the dark, on ice, according to the manufacturer’s recommendations. Respective IgG isotype controls were used to determine the nonspecific interactions between the antibodies and the cell surface proteins. Cells were washed twice with ice-cold PBS, re-suspended in 1% BSA-PBS flow cytometry buffer, and analyzed using a BD Accuri C6 flow cytometer. Surface receptor expression was determined by calculating the median fluorescence intensity of the target protein, minus the median fluorescence intensity of the corresponding isotype control. All data are shown relative to the corresponding monolayer-cultured samples of each cell-type.

### 4.5. Western Blotting

Immunoblotting analysis was performed, as previously described [24,37]. Cells lines were cultured in either monolayer or suspension growth conditions, and were harvested at specified time-points. Cells were washed with PBS and lysed using a radioimmunoprecipitation assay (RIPA) lysis buffer, with a protease inhibitor. Protein concentrations were determined using the bicinchoninic acid (BCA) protein assay (Pierce, Rockford, IL, USA). Equal amounts of lysis (20 μg) were resolved by electrophoresis, using 4–12% NuPAGE Bis-Tris gels and transferred to the PVDF membranes (Life Technologies, Carlsbad, CA, USA). Primary antibodies were used at the recommended manufacturer’s indications (1:500 to 1:1000). Membranes were stripped using the Restore Western Blot Stripping Buffer (Pierce) and re-probed with appropriate antibodies. Immunocomplexes were visualized with chemiluminescence using the Immobilon Western Chemiluminescent HRP Substrate. Antibodies specific to human DR4 (D9S1R), DR5 (D4E9), TNFR1 (C25C1), Caspase 3 (8G10), Caspase 8 (1C12), PARP (9542), and p62 (D5E2), were purchased from Cell Signaling Technologies (Danvers, MA, USA). Fas (C-20) was purchased from Santa Cruz Biotechnology (Dallas, TX, USA). GAPDH (2D4A7) and LC3B/MAP1LC3B (NB100) were purchased from Novus Biologicals (Littleton, CO, USA).

### 4.6. Gene Expression Analysis

Gene expression was analyzed using the Human OneArray^®^ Plus gene expression profiling service (HOA version 6.2, Phalanx Biotech Group, Inc., San Diego, CA, USA). RNA was extracted from the MDA-MB-231 and the MCF7 cell lines, cultured in monolayer, or in suspension culture, for 7 days. RNA quality was assessed using a NanoDrop ND-1000 with a pass criteria of absorbance ratios of A260/A280 ≥ 1.8 and A260/A230 ≥ 1.5. RNA integrity number (RIN) pass criteria of >6 was used to determine the acceptable RNA integrity. Gel electrophoresis was used to evaluate gDNA contamination. The data obtained were analyzed, using an Agilent 0.1 XDR Protocol. Gene expression fold changes were calculated by the Rosetta Resolver 7.2, with an error model adjusted by the Amersham Pairwise Ration Builder. Differential expression of genes was determined through the selection criteria of log2|fold change| ≥ 1 and *p* < 0.05. Data shown are the log2 ratios (suspension compared to monolayer) of each cell-type, with the corresponding *p*-value.

### 4.7. Imaging Flow Cytometry for Autophagic Flux Analysis

To determine the autophagy initiation under a suspension culture, multispectral imaging flow cytometry was performed. Cells were dissociated using Cell Stripper, washed once with ice-cold PBS, re-suspended, and fixed at a concentration of 10^6^ cells/mL in 4% formaldehyde for 10 min. Following fixation, cells were washed, twice, with PBS and then permeabilized using 0.01% Triton-X-100 in 1.0% BSA, in a PBS solution, for 5 min. Cells were stained as per the manufacturer’s recommendation with LC3-AF488 (Novus Biologicals, NB-60001384, Centennial, CO, USA), LAMP1-PE (Novus Biologicals, NBP2-25183), DR5-PE (R&D Systems, Minneapolis, MN, USA), in single fluorophore samples or combination samples, for the co-localization studies. In each experiment; 45,000 cells were acquired using a 12 channel Amnis^®^ FlowSight (EMD Millipore, Burlington, MA, USA) imaging flow cytometer, equipped with 405 nm and 488 nm lasers. Samples were acquired at 40× magnification. Single color controls were also acquired for compensation analysis. IDEAS^®^ (EMD Millipore) software was used for data collection and analysis.

Single cells were identified, using a bivariate plot of the cell area and aspect ratio from the bright field imaged population. In-focus cells were identified using the Gradient RMS of the bright field images, gating between 35.0 and 72.0 GRMS. Each sample was compensated using single-channel controls to minimize spectral spillover. The compensated files were analyzed for LC3 puncta formation and co-localization (bright detail similarity, BDS) on the double-positively-labeled populations. During the autophagosome formation, LC3 was expected to transition from a diffuse cytosolic LC3-I to a clustered autophagosome-membrane bound LC3-II. The LC3-puncta could then be used to find individual autophagosomes. Autophagosome formation was quantified, using a function mask to calculate the LC3-AF488 bright detail intensity spots (puncta), using a bright detail fluorescence intensity to cell fluorescence ratio of 4.0:1.0, to ensure only the autophagosomes are measured, without any noise from the cytoplasmic or non-specific staining. The BDI puncta were then counted, using the spot-counting feature and a spot radius of less than 1.0 pixel [52,53,54]. This radius was chosen to best distinguish the individual autophagosomes, as there is a heterogeneity in size and shape. Histograms were created for the frequency of cells with increasing autophagosome counts. Based on the basal levels of autophagy in the breast cancer cell lines cultured in the monolayer, cells with more than 7.0 LC3-AF488 puncta, per cell, were determined as undergoing increased levels of autophagy. The percentage of the cell population that were undergoing increased levels of autophagy were quantified. Autophagic flux was determined by the co-localization between the LC3-AF488 (autophagosome marker) and the LAMP1-PE (lysosomal marker). The co-localization was calculated using the Bright Detail Similarity (BDS) feature within the IDEAS software. The BDS feature calculates the pixel location of the bright detail intensities of both the LC3-AF488 and the LAMP1-PE. A Pearson’s correlation of the bright detail intensities’ pixel location was calculated, and log transformed into a BDS score, to determine the co-localization of two probes between the two channels. Cells undergoing autophagic flux were determined as having a BDS score above 1.8 (Pearson’s correlation of *r* ≈ 0.7). Co-localization between the LC3-AF488 and DR5-PE was determined, similarly.

### 4.8. Immunocytochemistry

For the immunocytochemical staining, the ZR75-1 cells were cultured in monolayer or suspension for 1, 3, and 7 days. Cells were dissociated with a non-enzymatic dissociation buffer, washed twice, and then re-suspended, at a concentration of 0.5 × 10^6^ cells/mL. Cells were spun onto the microscopy slides using a Thermoscientific Cytospin 4, at 1000 rpm, for three minutes. Cells were then fixed with 4% PFA for 10 min, washed twice, permeabilized with 0.01% Triton-X for 5 min at room temperature, washed twice, and then blocked using 3% BSA, in PBS, for 20 min. Samples were incubated with primary antibodies, overnight, in a blocking solution, at 4 °C. The following day, the slides were washed, twice, with PBS and incubated for an hour with the corresponding conjugated secondary antibodies (goat anti-mouse-AF594 (Life Technologies, A11012) and goat anti-rabbit-AF488 (Life Technologies, A10667). Samples were washed twice and mounted with the DAPI-slow fade mounting solution (Vector Laboratories, H-1200, Burlingame, CA, USA). Confocal imaging was performed using a Zeiss LSM880 (Zeiss, Thornwood, NY, USA). Images were obtained at 40× magnification and analyzed using Zen software (Zeiss). The negative controls were the unlabeled samples and samples stained with only the secondary antibodies.

### 4.9. Statistical Analysis

Statistical analysis for cell viability, flow cytometry, western blotting, and co-localization studies were performed using the GraphPad Prism 6 (GraphPad Software, San Diego, CA, USA). Statistical comparisons were determined using a one-way ANOVA and an unpaired t-test, with a Welch’s correction. All assays were completed with an *N* ≥ 3. Statistical significance has been shown as either * *p* < 0.05 or ** *p* < 0.01, as indicated in the results.

## 5. Conclusions

Our data suggest that CTCs may evade the TNF cytokine-mediated immune surveillance through downregulation of the DR expression, thereby, contributing to tumor metastasis or chemoresistance. 

## Figures and Tables

**Figure 1 cancers-11-00094-f001:**
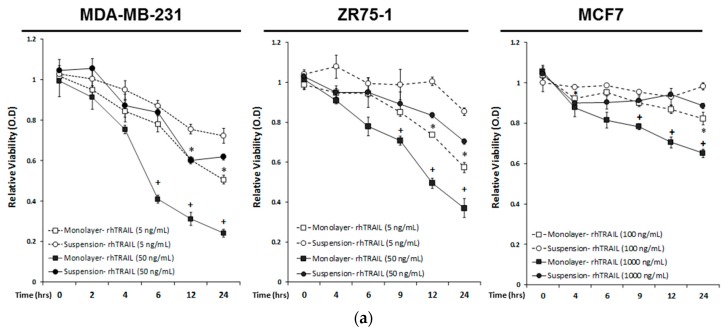
Breast cancer cells cultured under the suspension condition acquire resistance to recombinant human TNF-related apoptosis inducing ligand (rhTRAIL)-induced apoptosis. (**a**) The indicated breast cancer cell lines were cultured under monolayer adherent or non-adherent suspension conditions (see details in Materials and Methods section). Cells were seeded at 10,000 cells per well and were then treated with the rhTRAIL (5 ng/mL and 50 ng/mL for MDA-MB-231 and ZR75-1 cell lines; 100 ng/mL and 1000 ng/mL for MCF7 cell lines reflecting the previously determined IC50 to rhTRAIL treatment [37]), over 24 h. Relative viability was measured at hour intervals, using an MTT assay, and was normalized to the non-treated controls. Values are means ± SEM of triplicates. (* *p* < 0.05 monolayer culture relative to suspension at same time point with rhTRAIL treatment of 5 ng/mL for MDA-MB-231 and ZR75-1 or 100 ng/mL for MCF7 cells; ^+^
*p* < 0.05 monolayer culture relative to suspension at same time point with rhTRAIL treatment of 50 ng/mL for MDA-MB-231 and ZR75-1, or 1000 ng/mL for MCF7 cells; *n* = 3). (**b**) Western blot analysis of caspase and PARP cleavage following the rhTRAIL treatment.

**Figure 2 cancers-11-00094-f002:**
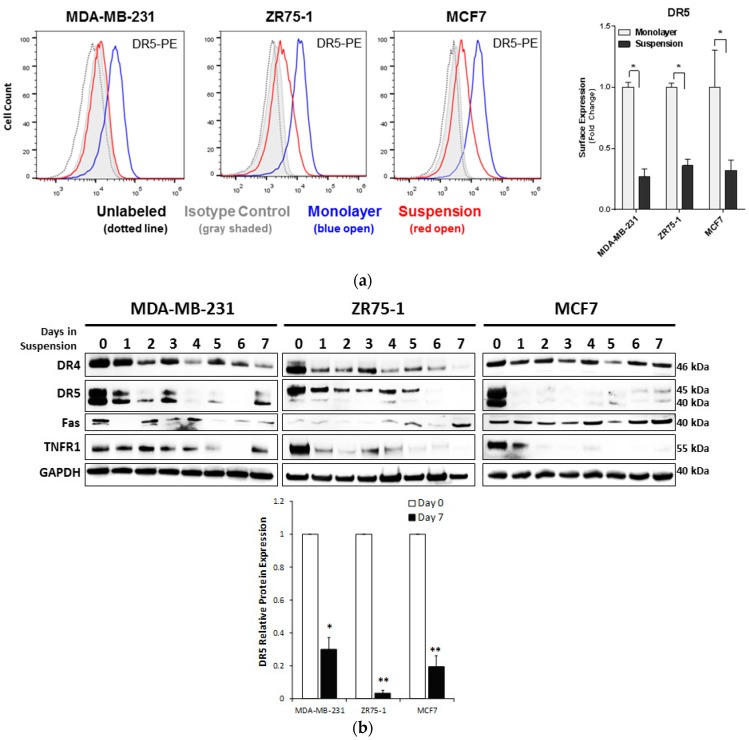
Surface expression of death receptors are decreased in breast cancer cells under suspension conditions. (**a**) Surface expression of the DR5 was analyzed by flow cytometry on breast cancer cells (BCCs) cultured in monolayer and suspension, for seven days. Surface expression was analyzed using Relative Median Fluorescence Intensity (RMFI) of the DR5, determined by the Median Fluorescence Intensity (MFI) obtained for the suspension-cultured cells, normalized to the corresponding MFI obtained for monolayer cells (Day 0) (mean ± SEM; * *p* < 0.05; *n* = 3). The reduction in surface level protein is indicated with a median shift of the suspension cultured cells (red line), from the monolayer (blue line) towards the isotype (gray shading), and the unlabeled control (dotted line). (**b**) Western blot analysis of the BCC lines cultured in the suspension condition and collected each day. Cell lysates were analyzed for DR4, DR5, Fas, and TNFR1 with GAPDH as a loading control (quantification of each blot can be found in Appendix A). Relative protein expression of the DR5 (relative densitometric analysis) to monolayer (day 0). (* *p* < 0.05, ** *p* < 0.01 to monolayer; *n* = 3).

**Figure 3 cancers-11-00094-f003:**
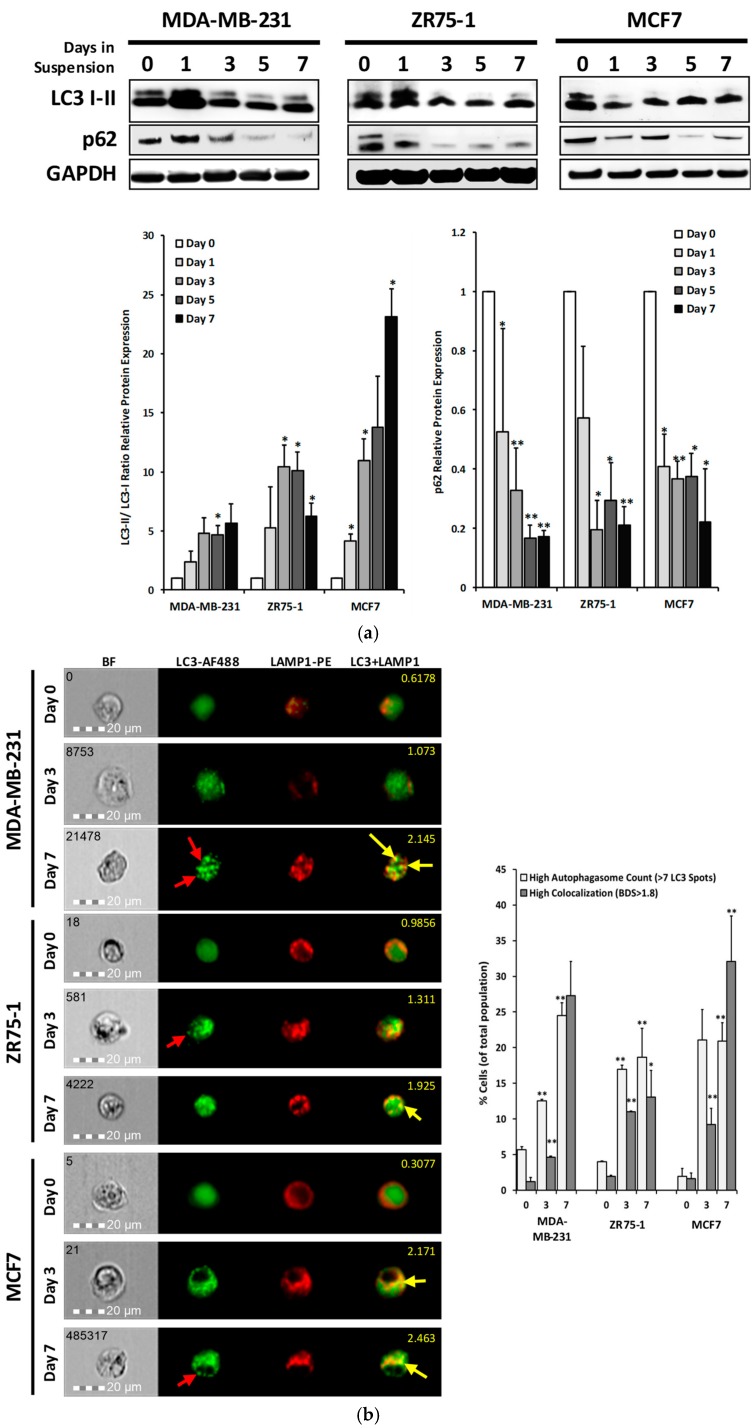
Non-adherent cultured cells increased the autophagic flux as measured by the LC3 I-II turnover and autolysosomal formation. (**a**) Western blot analysis of the BCC lines, cultured in monolayer or suspension condition up to seven days. Cell lysates were analyzed for LC3-I (top band) and LC3-II (bottom band), p62, and GAPDH. LC3-II to LC3-I relative protein ratio indicating the LC3 turnover compared to the monolayer (day 0) (mean ± SEM; *n* = 3; * *p* < 0.05). Relative expression of the p62 to the monolayer-cultured cells (mean ± SEM; *n* = 3; * *p* < 0.05). (**b**) Images of the MDA-MB-231, ZR75-1 and the MCF7 cells cultured in monolayer (day 0) or suspension for three or seven days, captured using imaging flow cytometry. Figure shows brightfield (BF), LC3-AF488 (green), LAMP1-PE (red), and a composite image of the co-localization of LC3 and LAMP1 (yellow). Autophagosomes are indicated with red arrows and the co-localization of LC3 and LAMP1, representing the autolysosome formation, is shown with yellow arrows. Corresponding bright detail similarity score between the LC3-AF488 and the LAMP1-PE puncta is shown in yellow. (Single fluorophore controls are provided in Appendix A). Population analysis of cells undergoing autophagy (high autophagosome counts (≥ 7 LC3+ spots per cell)) and autophagic flux (high co-localization of LC3-AF488 and LAMP1-PE (bright detail similarity score ≥ 1.8)) (mean ± SEM; *n* = 3; * *p* < 0.05, ** *p* < 0.01 relative to monolayer).

**Figure 4 cancers-11-00094-f004:**
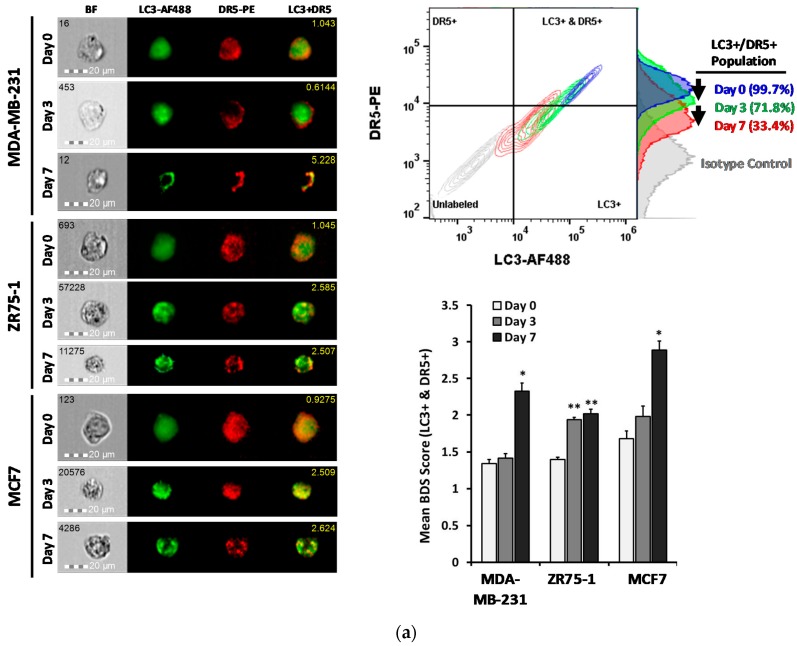
DR5 localized to autophagosome for degradation under suspension culture. (**a**) Representative images of the MDA-MB-231, ZR75-1 and MCF7 cells, cultured in monolayer (Day 0) or suspension for three or seven days, captured using imaging flow cytometry. Figure shows brightfield (BF), LC3-AF488 (green), DR5-PE (red), and a composite image of the co-localization of the LC3 and the DR5 (yellow), with the associated bright detail similarity score, per cell (yellow number in the top right). (Single fluorophore controls are shown in Appendix A). Bivariate flow cytometry plots of the LC3-AF488 and the DR5-PE, with the associated DR5-PE histogram, per time-point. Population percentages are shown for the dual-positive population. Mean bright detail similarity scores of the breast cancer cells cultured in monolayer or suspension culture for three or seven days. Bright detail similarity (BDS) scores were calculated on the dual-positive populations only (mean ± SEM; *n* = 3; * *p* < 0.05, ** *p* < 0.01 relative to monolayer). (**b**) Immunocytochemistry and confocal microscopy of the cyto-spun ZR75-1 cells cultured in monolayer (day 0) or suspension for three or seven days. LC3 (AF488, green), DR5 (AF594, red), DAPI, brightfield, and composite micrographs are shown. Yellow arrows indicate co-localized LC3 and DR5. Images taken at 40×.

**Table 1 cancers-11-00094-t001:** Relative gene expression (log_2_ ratio (day 7 suspension/monolayer culture) of the DR4, DR5, Fas, and TNFR1, determined by the Phalanx Biotech Human OneArray DNA microarray (data shown as log_2_ ratio (*p*-value)).

Protein	Gene ID	MDA-MB-231 Day 7/Day 0	MCF7 Day 7/Day 0
DR4	TNFRSF10a	−0.711 (0.034)	0.066 (0.835)
DR5	TNFRSF10b	0.281 (0.260)	0.302 (0.386)
Fas	FAS	0.408 (0.293)	0.029 (0.976)
TNFR1	TNFRSF1A	0.151 (0.530)	0.220 (0.188)

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
