# Peer review of "Circulating Tumor Cells Develop Resistance to TRAIL-Induced Apoptosis Through Autophagic Removal of Death Receptor 5: Evidence from an In Vitro Model"

_cancers, 2019, doi:10.3390/cancers11010094_

Reviewer 1 Report

Twomey and Zhang have utilised an in vitro model of breast cancer to investigate resistance to TRAIL-induced apoptosis. They did this by culturing breast cancer cells under non-adherent conditions, and analysed TRAIL induced apoptosis via the caspase cascade and by localisation of death receptors to autophagosomes. Overall, this is a neat study with, overall, very convincing data. I would recommend acceptance after the following suggestions/questions are answered:

1)      Introduction, line 27 – CTCs are not only shed from the primary tumour, they are also shed from metastatic lesions. Please add this.

2)      Please add P values to the text (page 3) regarding the viability assays.

3)      In Figure 1a, please state why a 20-fold higher concentration of rhTRAIL was needed for MCF7 cells to aid the non-specialist reader.

4)      In Figure 1b, why are no bands shown for 9 and 12 hrs for MDA-MB-231s?

5)      In Figure 1b, why are no bands shown for 24 hrs in any of the cell lines? Looking at Figure 1a there are clearly many viable cells.

6)      In Figure 1b, the ~17 kDa band representing cleaved Caspase 3 is not clear in any of the blots. This suggests 2 things: 1) the antibody used is not specific enough for cleaved Caspase 3 or 2) PARP cleavage is occurring via a Caspase 3 independent mechanism. Have the authors considered analysing Caspase 6 or 7?

7)      Although MCF7s are Caspase 3 deficient, this is not made clear and should be stated either in the results text or in Figure 1b so that non-specialist readers understand why no blot for this caspase is given. This further enhances the theory that PARP cleavage occurs through caspase 6 in these cells. Indeed, Zheng et al. (2000) Nat Med. 2000;6:1241–1247 have suggested this.

8)      In Figure 2b, the upper band in DR5 is predominant in ZR75-1 cells. Is there a particular reason for this?

9)      In Figure 2b, please add bands sizes in each blot.

10)   It would be useful to see blots for the expression of each protein in monolayer cultures.

11)   Line 261. Please change the statement “we have shown autophagic flux as a mechanism through which tumour cells internalize and degrade death receptors when they are shed from the primary tumour and circulate in the blood stream”. This study used and in vitro model to suggest this, but can only be confirmed by being validated in circulating tumour cells obtained from breast cancer patients.

Author Response

The authors have addressed Reviewer 1's comments and have made the recommended changes.

Reviewer 2 Report

In this paper the authors report on experiments using an artificial cell model for circulating tumor cells. These cells became more resistant to TRAIL induced cell death when compared to adherent cells. This correlated with Death Receptor 5 (DR5) abundance and its localization to autophagosomes suggested lysosomal degradation as mechanism for resistance and thus allowing CTCs to survive unattached in the presence of TRAIL.

The paper is generally well written and most of the methods are sufficiently described.

I have only a few points that should be improved or explained:

The cell model relies on the application of culture plastic that does not support adhesion. In several studies, such material has been used to obtain 3D-spheroids; especially in MCF-7 this is working effectively and also MDA-MB-231 was described to form aggregates. Please comment on the amount of cells that attach to each other and provide some photographs of such cultures!

I think it would strengthen the conclusions of this paper to perform additional experiments: DR5-receptor is supposed to be downregulated post transcriptionally by autophagy, so an inhibitor of this process should be able to increase DR5 protein back to ”normal” and render the cells sensitive to TRAIL again. Has this been tried by the authors? Would it be feasible?

Minor points:

In methods the MTT concentration is missing.

For some centrifugations x g is not given.

How were the array gene expression raw data normalized, is this part of the described software? I am not familiar with the Agilent 0.1 XDR Protocol or the Rosetta, so it might be better to explain this further.

Blocking immunohistochemistry by 3% BSA: dissolved in …?

Is the spatial resolution of the microscope used at these parameters known?

Author Response

Round  2

Reviewer 2 Report

The authors provided a revised version of their manuscript accompanying a point by point rebuttal. This revised version has been improved according to my comments and can now be published without further changes.

Author Response

The author acknowledges that no additional changes are recommended.